# InstructExcel: A Benchmark for Natural Language Instruction in Excel

**Justin Payan**[2*]   **Swaroop Mishra**[3*†]   **Mukul Singh**[1]   **Carina Negreanu**[1]
**Christian Poelitz**[1]   **Chitta Baral**[3]   **Subhro Roy**[1]   **Rasika Chakravarthy**[1]
**Benjamin Van Durme**[1]   **Elnaz Nouri**[1]
[1]Microsoft, [2]UMass Amherst, [3]Arizona State University

## Abstract

With the evolution of Large Language Models (LLMs) we can solve increasingly more complex NLP tasks across various domains, including spreadsheets. This work investigates whether LLMs can generate code (Excel Office-Scripts, a TypeScript API for executing many tasks in Excel) that solves Excel specific tasks provided via natural language user instructions. To do so we introduce a new large-scale benchmark, INSTRUCTEXCEL,[1] created by leveraging the 'Automate' feature in Excel to automatically generate OfficeScripts from users' actions. Our benchmark includes over 10k samples covering 170+ Excel operations across 2,000 publicly available Excel spreadsheets. Experiments across various zero-shot and few-shot settings show that INSTRUCTEXCEL is a hard benchmark for state of the art models like GPT-4. We observe that (1) using GPT-4 over GPT-3.5, (2) providing more in-context examples, and (3) dynamic prompting can help improve performance on this benchmark.

## 1 Introduction

Spreadsheet software, especially Microsoft Excel, is used every day by a diverse set of users (from complete novices to highly sophisticated power users) to solve a wide range of important tasks. To help users accomplish their tasks, Excel has introduced an array of features, including Analyze Data[2] where users can specify their task in natural language (NL) for a limited scope of tasks. As even seemingly simple manipulations can often require knowledge of complex concepts like loops, conditionals, and regular expressions (Desai et al.,

2016) it is important to further diversify the range of tasks AI assistants can support.

Meanwhile, the advent of large language models, together with the instruction learning paradigm (Mishra et al., 2022b; Wei et al., 2022; Ouyang et al., 2022; Sanh et al., 2022), has paved the way for non-experts to accomplish many tasks just by providing instructions in natural language. *Can we leverage this novel paradigm to let Excel users automatically execute complex tasks from NL instructions?*

Excel OfficeScripts[3] provide a versatile Type-Script API for most standard functions in Excel, giving us a simple intermediate representation for executing NL instructions in Excel. Our task is to convert a user's description of a simple action or set of actions in Excel (such as editing formatting) into an executable OfficeScript accomplishing the action(s) on the given spreadsheet. Figure 1 shows an example input and output, with additional examples in Table 5 of Appendix A.

We introduce INSTRUCTEXCEL, a benchmark to investigate the instruction paradigm for generating OfficeScripts from NL instructions. We extract publicly available Excel sheets and ask crowdworkers to think of an action they want to perform on the sheet and write a summary of it in NL. Subsequently, we ask crowdworkers to execute that action in the sheet and use the Automate feature to record the action and generate the underlying code. Automatic code generation improves output quality and eliminates potential bugs prevalent in human-written code. INSTRUCTEXCEL contains over 10k samples covering 170+ OfficeScripts operations across 2,000 publicly available Excel sheets.

We evaluate OpenAI's GPT-3.5 Turbo and GPT-4 models (OpenAI, 2023) as well as a T5 model

---

*  Work done during internship at Microsoft

†  Currently at Google DeepMind

[1]INSTRUCTEXCEL, along with the code used in our experiments, is available at `https://github.com/microsoft/InstructExcel`.

[2]https://support.microsoft.com/en-us/office/get-insights-with-analyze-data-aa105149-1e48-446d-b3df-872dff70a866

---

[3]OfficeScripts API documentation: `https://learn.microsoft.com/en-us/javascript/api/office-scripts/excelscript?view=office-scripts`.

(Raffel et al., 2020), on INSTRUCTEXCEL and find it to be a challenging benchmark even using state of the art tools, indicating significant room for future work on this task. We also find that GPT-4 improves over GPT-3.5, and dynamic prompting and more in-context examples improve performance, but additional in-context instructions do not help.

In addition to evaluating capabilities of state-of-the-art models on the full benchmark, we also illustrate the broader utility of INSTRUCTEXCEL through a case study on conditional formatting rules. INSTRUCTEXCEL contains 660 examples that represent conditional formatting tasks, which are studied in isolation (Singh et al., 2022). Viewed through the lens of this particular task, INSTRUCTEXCEL provides a data source that can be further processed to enable experiments with task-specific ground truth. Many other sub-tasks can be extracted from INSTRUCTEXCEL in the same manner, such as charting/plotting (Lee et al., 2021) or formula creation (Chen et al., 2021b).

We believe INSTRUCTEXCEL will encourage development of user-friendly methods to automate various Excel tasks and help non-expert Excel users in their daily workflow.

## 2 Related Work

### 2.1 Instruction Learning Paradigm

The instruction paradigm (Mishra et al., 2022b; Wei et al., 2022; Sanh et al., 2022; Ouyang et al., 2022) provides a user-friendly option to leverage ML models just by providing instructions without requiring underlying technical knowledge. Instructions describe tasks in NL (Efrat and Levy, 2020; Weller et al., 2020), and are shown to help models generalize to unseen tasks (Mishra et al., 2022b; Wei et al., 2022; Ouyang et al., 2022; Wang et al., 2022b; Xu et al., 2023; Zhong et al., 2021; Gupta et al., 2023; Patel et al., 2022; Puri et al., 2023; Mishra and Nouri, 2022; Chung et al., 2022). Recent developments in large language models (Brown et al., 2020; Chowdhery et al., 2022) have led to the successful use of instruction learning methods in various applications, such as dialog (Gupta et al., 2022), tabular question answering (Luo et al., 2022), relation extraction (Chen et al., 2021a), biomedical applications (Parmar et al., 2022), NER (Wang et al., 2022a), program synthesis (Kuznia et al., 2022), and style transfer (Reif et al., 2022). Natural Instructions (Mishra et al., 2022b), Supernatural Instructions (Wang et al., 2022c), and Promptsource (Bach et al., 2022) are some popular instruction learning benchmarks, however they are focused on general NLP. In contrast to prior works, we focus on the application of the instruction paradigm in the Excel domain.

### 2.2 Program Synthesis

The recent success of large language models (LLMs) like Codex (Chen et al., 2021), GPT4 (OpenAI, 2023), PaLM (Chowdhery et al., 2022), PaLM2 (Anil et al., 2023), StarCoder (Li et al., 2023) and WizardCoder (Luo et al., 2023) has led to significant progress in program synthesis. LLMs have been found to generalize to many different code generation tasks. The models vary in size and training schemes and their success rate in performing code generation tasks (Xu et al., 2022). Several works leverage pre-trained models to map NL queries to sequences of API elements or other intermediate representations, which can be translated into code, though this may not be necessary with models that have been pre-trained on code (Hadi et al., 2022; Shin and Van Durme, 2022). They have also been used for in-place data transformations (Narayan et al., 2022). Pre-trained models have proven to be especially strong when combined with clever post-processing steps or constrained decoding, for example in Jigsaw (Jain et al., 2022), Synchromesh (Poesia et al., 2022), and PICARD (Scholak et al., 2021). Chan et al. (2022) investigates the impact of training on a large number of different tasks. Other important aspects studied include length generalization (Anil et al., 2022), compositional generalization (Shi et al., 2022), reverse engineering (Pearce et al., 2022), and generating development tools (Bareiß et al., 2022). The task of NL to Code is broadly of interest to the semantic parsing literature (Kamath and Das, 2018; Zhong et al., 2022; Zhao et al., 2022).

Existing benchmarks for automated code generation include CoNaLa (Yin et al., 2018), DJANGO (Oda et al., 2015), HumanEval (Chen et al., 2021), MBPP and MathQA-Python (Austin et al., 2021), APPS (Hendrycks et al., 2021), and CodeContests (Li et al., 2022). In contrast to these benchmarks that target general-purpose programming languages, our work INSTRUCTEXCEL instead focuses on a specialized Excel API, which LLMs are less likely to have encountered during pre-training.

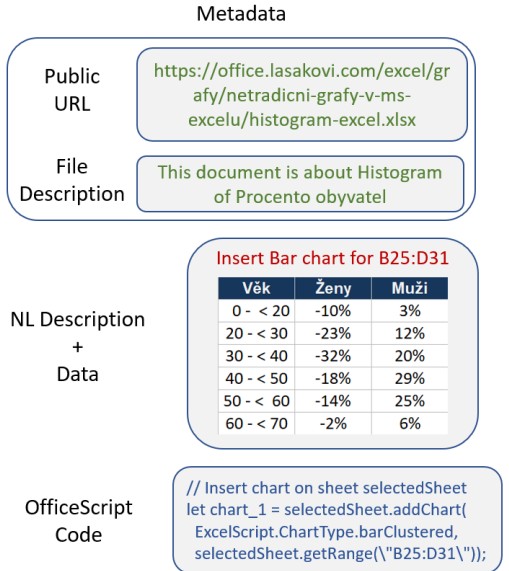

Figure 1: Schema of INSTRUCTEXCEL: A sample input-output pair in INSTRUCTEXCEL. A natural language description and the linearized spreadsheet data are the inputs and the Excel OfficeScript code is the desired output. We also have two additional pieces of metadata, the URL for the Excel file and a description of the file's contents.

## 3  INSTRUCTEXCEL

In this section, we describe the INSTRUCTEXCEL benchmark. We start with the schema used to represent the task, followed by our dataset construction procedure, and an overview of the dataset statistics.

### 3.1  Task

The user inputs actions – such as clicking buttons, writing formulas, and inserting columns. These actions correspond to OfficeScript code which produces an equivalent effect on the Excel spreadsheet. In addition, the user describes the intended result of their actions using a natural language description. Our task is to map the natural language description input, along with the contents of the spreadsheet, to the OfficeScript code output.

### 3.2  Schema

Figure 1 shows the schema of INSTRUCTEXCEL, along with an example. It has 4 elements: input, output, Excel file URL and Excel file description (see Table 5 in Appendix A for more examples).

**Input**  Each input contains a natural language instruction that specifies the desired manipulation of the Excel spreadsheet, e.g. "highlight the 1st row of sheet 1." This can also contain references to cells

and other Excel-specific details associated with the sheet. The input is typically a single line of text provided by the crowdworker.

**Output**  The output field contains the OfficeScript code generated when the user records the actions taken to accomplish their task. OfficeScripts are written in TypeScript, a superset of JavaScript.

Each script must contain a main function with the ExcelScript.Workbook type as its first parameter. When the function runs, the Excel application invokes the main function by providing the workbook as its first parameter.

**Excel File Name**  Each entry includes a link pointing to the Excel file. Although we do not release the spreadsheets directly as part of the benchmark, each spreadsheet can be downloaded from its associated URL.

**Excel File Description**  Since some Excel files are very large and may not fit within the limited context size of smaller language models, we have another field where the crowdworker describes the contents of the Excel file in natural language. This is usually one or two lines of text.

### 3.3  Constructing INSTRUCTEXCEL

**Selecting Excel Files**  Starting from a dataset of 5,000 Excel documents from publicly available links on the web, we removed any files that are in languages other than English or that were password-protected, corrupted, or otherwise inaccessible. We also applied a size criteria (20 KB < filesize < 30 KB) to avoid Excel files that were too small to have meaningful data or too large for human annotators to understand and study in the provided time. We eventually selected a final set of 2,000 Excel documents to be further annotated.

**Data Creation Process**  We used crowdsourcing to record actions over the set of Excel documents. We used a crowdsourcing platform called the Universal Human Relevance System.[4] We recruited English speaking annotators from India. We used an intermediate third party vendor which assures quality of the tasks through management of communication and training of the annotators. We requested our vendor to arrange a pool of annotators who have basic familiarity with Excel applications. We also provided the vendor a list of popular Excel

---

[4]UHRS located at: https://prod.uhrs.playmsn.com/UHRS/.

operations which they must use. We ensured that our third party vendor had access to the Automate feature in Excel Web version and that they could access the Excel files of our dataset through the Excel Web version.

For each document the user was instructed to input a natural language description of the file. They were then asked to type a natural language instruction, record themselves performing the relevant action for the instruction, and copy-paste the resulting code generated by the Automate feature in Excel Web. They repeated that process 5 times per spreadsheet. The full data collection process is illustrated in Figure 2. A screenshot of the interface for the human intelligence task (HIT) is provided in Appendix B.

**Qualification, Crowdworker Compensation and Data Quality Check**  We provided 2 rounds of pilot HITs to the vendor's annotators, reviewed their responses, and made clarifications to the instructions based on the feedback. Specifically, we instructed annotators not to overuse or underuse certain actions after observing the action distribution in pilot rounds. We provided various examples of queries which could be toggled on and off within the interface. We assessed the qualification of the crowdworkers based on the responses to the pilot HITs submitted to us. In addition to the hourly rate of 12 USD for the annotation work, we also covered the vendor's management and training fees to assure response quality.

**Consent and Privacy Control**  Annotators had the choice of performing the requested tasks only after signing a consent form. In addition, we provided explicit instructions in the HITs that disallowed sharing of any personal and identifiable information when providing responses.

### 3.4 Statistics

Table 1 shows key statistics of INSTRUCTEXCEL. The most common words used in the Excel sheet descriptions are 'year', 'name', 'form', 'calendar', 'table', and 'content.' These words represent common Excel workflows. 'Formula', 'background', 'color', 'conditional', and 'formatting' are the most common words in the NL instructions.

Figure 3 shows the most common methods used in the recorded OfficeScripts. We see a fairly broad distribution over important Excel functionalities. The top methods correspond with the most common words used in the users' instructions – for ex-

| Category | # of instances |
|---|---|
| Samples | 10347 |
| Excel files | 2000 |
| Operations | 177 |
| Avg. Ops./Sample | 8.9 |

Table 1: Statistics of INSTRUCTEXCEL. Operations are distinct API methods that represent various Excel tasks.

| EM | ROUGE | F1 | BLEU |
|---|---|---|---|
| 21.30 | 67.64 | 65.23 | 60.45 |

Table 2: Metrics measured between pairs of code outputs belonging to the same natural language queries. The relatively low exact match but relatively high ROUGE, F1, and BLEU scores reflect the fact that some queries have multiple solutions that vary only slightly.

ample, use of the word 'formula' corresponds with the method call 'setFormulaLocal.' Many of the top methods require only one parameter. However, some of the methods require fairly complex parameters. The 'setFormulaLocal' method requires formulas input as strings, and 'setRule' requires complicated conditional format rule objects that must be constructed before being passed to the method call. Some common actions performed by users require sequences of multiple method calls; for instance, inserting a shape requires specifying the location, type, and color of the shape.

We found 76 queries that were repeated at least once in the benchmark. Most of these queries are general queries such as "freeze row 1" or "sort column G." We compute the exact match, ROUGE, F1, and SacreBLEU scores between all pairs of code blocks belonging to the same natural language user queries, displayed in Table 2. Although the repeated queries often have only one correct code solution (for example, "freeze row 1" has only one solution), many queries can be satisfied with multiple code solutions (i.e., "Create chart on worksheet Sheet1" can be satisfied with any chart type).

## 4 Experiment

We conduct experiments with supervised, zero-shot and few-shot learning to assess the performance of popular LLMs on INSTRUCTEXCEL. We use GPT-3.5 Turbo and GPT-4 in our experiments (OpenAI, 2023), specifically the "gpt-3.5-turbo-16k"

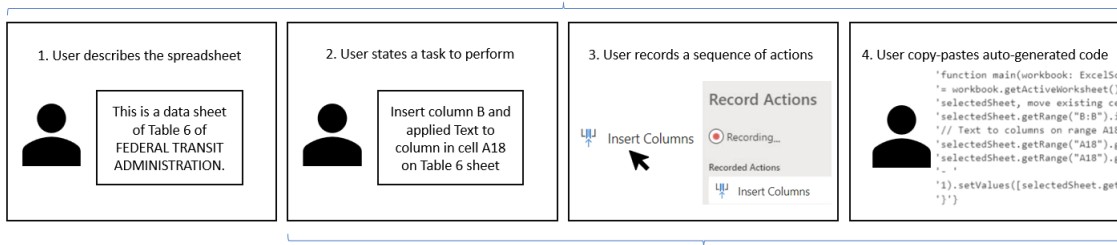

Figure 2: Steps in data collection.

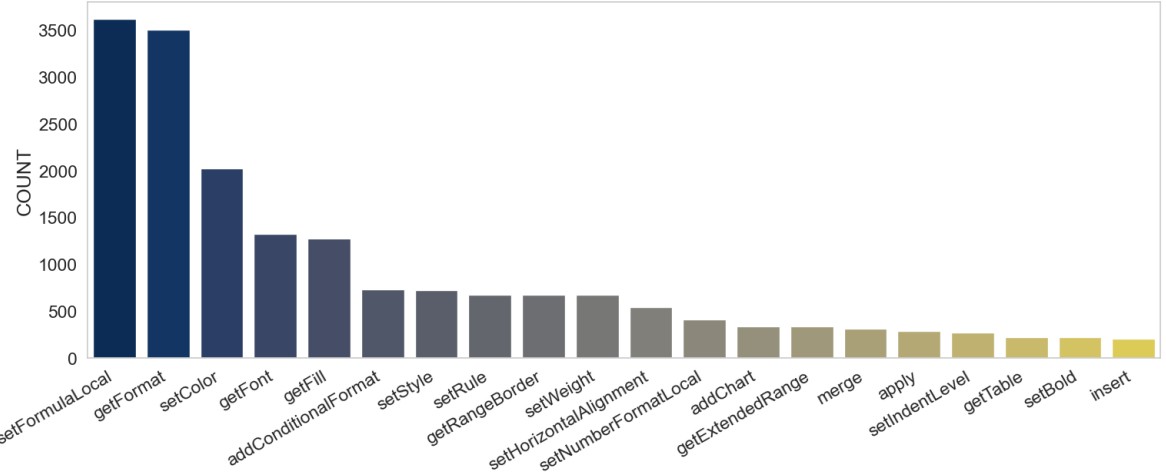

Figure 3: Plot showing the distribution of the top 20 most-common Excel operations in our dataset. The X axis shows the unique operations, and the Y axis shows the frequency of the operations in INSTRUCTEXCEL.

and "gpt-4-32k" models.[5] We also experiment with finetuning the T5 Large LM Adapt model (Raffel et al., 2020).[6]

In all experiments, a single example consists of a triplet with 1) a natural language query, 2) the data from the spreadsheet converted into a string using a Pandas ExcelFile object, and 3) the ground truth or generated OfficeScripts code. We elaborate below on our full experiment setup.

## 4.1 Data Splits

We exclude examples from the benchmark that have broken URLs for the purposes of our experiments, as some of the URLs have expired since data collection and annotation occurred.[7] We divide the remaining examples into train, dev and test splits containing 4033, 1000 and 1000 instances

[5] https://platform.openai.com/docs/models
[6] https://huggingface.co/google/t5-large-lm-adapt.
[7] We intend to continue to collect and provide additional data in our public repository, so some natural URL expiry will not impact the utility of the dataset.

respectively. Considering the computational cost and rate limit associated with GPT usage, we also form a test-set subset of 200 samples which we use in our experiments.

## 4.2 Setup

### 4.2.1 Zero-shot

In this setting we do not provide any examples to the model. Instead, we provide the prompt 'Generate a function with excel script to execute the action given below in NL.' In order to make the task easier, we also add the common boilerplate code which starts every OfficeScript to the prompt as exemplified in Figure 4. The output is limited to 256 tokens, and we impose this limit in all other settings as well.

### 4.2.2 Few-shot and Max-shot

We use the in-context learning setup (Brown et al., 2020), showing a few examples before prompting the model to respond to the input query. We experiment with two different few-shot learning setups. First, we evaluate models in 3-shot learning (3 ex-

```
Excel Script Function:
function main (workbook: ExcelScript.Workbook) {
        let selectedSheet = workbook.getActiveWorksheet();
        //
```

Figure 4: Standard code template included at the end of each prompt.

amples are presented), where in-context examples are included after the instructions but before the query. Second, we evaluate models in the Max-shot setup, in which we add 10 examples. Unless otherwise specified, the examples used to prompt the model are randomly chosen and put in random order ahead of time, and we use the same examples for all test instances.

### 4.2.3 Max-shot+Instruction

Motivated by the success of detailed instructions in improving model performance (Mishra et al., 2022b; Wei et al., 2022; Ouyang et al., 2022; Mishra et al., 2022a), we prompt with a longer NL instruction along with 10 examples. The longer instruction is listed in Appendix C.

### 4.2.4 Finetuning

We finetune only the T5 model, as finetuning GPT-3.5 Turbo and GPT-4 is costly. Details of finetuning and the parameters for T5 inference are listed in Appendix D.

### 4.3 Data Inclusion

Although we try to include as much data as possible in the prompts, the spreadsheets can often be too large to fit all of them in the prompt. This problem is especially pronounced in the Max-shot setting. We first calculate the number of tokens in the prompt when using all available data for each in-context example and the query. If the number of tokens exceeds the context length for the model, we remove the second half of each data string. We repeat this process until the prompt is small enough to fit in context, or all data has been deleted. If all data has been deleted and the prompt is still too large, we revert to the full data for each example and simply truncate the string of in-context examples to make the prompt fit in context.

To understand the impact of data truncation, we analyze the two most space-restricted settings in our experiments, the Max-shot+Instruction settings with both the GPT3.5 model with 16k context size, and the GPT4 model with 32k context size + API

in the prompt. Both settings always include the intended 10 in-context examples in the prompt, and the data included in the examples retain an average of 19.36 lines (for GPT-3.5) and 15.36 lines (for GPT4+API). This typically guarantees that the headers and some data are included for each of the in-context examples and the query. We hypothesize that increasing context window size will result in performance improvements.

### 4.4 API in Context

To inform GPT of the API, we experiment with appending the API to every prompt. The API consists of an alphabetically sorted list of all classes, enums, methods, and properties in the OfficeScripts API, along with the accepted parameters for methods and the fields of all enums. We append the API immediately under the instruction in the prompt, before the in-context examples and the query.

### 4.5 Dynamic Prompting

One other method which can inform GPT of the relevant API elements for a given input is dynamic prompting (Liu et al., 2022). For dynamic prompting, given a natural language input, we search for the most similar natural language inputs in the training set, measured according to F1 score of the normalized text. We then append the top 3 or 10 examples from the training set, including the linearized data from each example. This prompting strategy not only provides in-context examples that are semantically similar to the query, but also increases the likelihood that the correct elements of the API are directly exposed in the context.

### 4.6 Evaluation

We rely on textual similarity-based evaluation metrics that can easily be reused: Exact Match (EM) comparing the normalized code prediction with the gold standard, F1, ROUGE (Lin, 2004) and SacreBLEU (Post, 2018) to estimate the textual similarity between predicted code and the gold standard. We use the implementations of ROUGE and SacreBLEU from HuggingFace Datasets,[8] and we use the built-in stemmer and ROUGE-L metric for ROUGE. We remove the boilerplate code (shown in Figure 4) before evaluation.

We also require a metric that measures semantic equivalence of code outputs without penalizing arbitrary choices like variable names or choices for

---

[8]https://huggingface.co/docs/datasets/index.

arbitrary parameter values. Some parameters and numbers cannot be directly inferred from the natural language query or the spreadsheet data, and are sometimes (but not always) arbitrary choices that do not impact the correctness of the code. Queries such as "insert shape in sheet December 2021" or "Change the legends position on sheet data in chart" can be correctly satisfied using many different parameter values for the shape type and legend position, while queries such as "Apply group to A7:A11 on sheet1" require using the specific parameter values A7:A11. Distinguishing these cases is a non-trivial task, and some use-cases are more parameter-sensitive than others. In Section 6, we analyze the subset of INSTRUCTEXCEL corresponding to conditional formatting operators, which admit evaluation by execution equivalence. However, evaluating relevance to general NL queries automatically is far more challenging.

To circumvent the general question of semantic equivalence, in addition to computing metrics on the original predicted and gold standard code, we also report the value for each metric when ranges, numbers, and parameters are stripped from the examples and replaced with placeholder values. We call this *function-based evaluation*, since this approach enables us to study functional equivalence of the output programs while ignoring some details.

We also annotate all predicted outputs from the three models with the highest performance on automated metrics. For each predicted code output, we give a binary judgment on whether the predicted code output satisfies the request given by the user. To aid in annotation, we perform our annotations while looking at the gold standard outputs, and we consult the spreadsheets when data from the spreadsheet is necessary to judge correctness.

## 5  Results

Table 3 shows all metrics across various modeling setups, but only for Max-shot prompting. We find that Max-shot outperforms Zero-shot, Few-shot, and Max-shot+Instruction (see full results in Appendix E) across all models, so we only report Max-shot in-context learning and finetuning results in this section. We summarize our main findings below.

**Finetuning Improves over In-context Learning on Automated Metrics**  Model performance after finetuning is notably higher than model performance in all other settings, when comparing on

automated metrics. However, GPT-4+DP outperforms the finetuned T5 model when comparing manual annotation score. The relative performance gain of finetuning over in-context learning is therefore unclear, though both appear effective.

**GPT-4 outperforms GPT-3.5 Turbo**  For the Few-shot (3), Max-shot and Max-shot+Instruction settings, we observe that the performance of GPT-4 is higher than GPT-3.5 Turbo (details in Appendix E).

**More In-context Examples Help Few-Shot Capabilities**  Across both GPT models, we observe that the Max-shot model performance is much higher than the Few-shot and Zero-shot performance (see Appendix E for details). This potentially indicates that addition of in-context examples can greatly help LLMs generate OfficeScript code.

**Detailed Instructions Do Not Help**  The Max-shot+Instruction baseline is not remarkably different from the Max-shot baseline. Considering the sensitivity of model performance to instruction framing (Mishra et al., 2022a; Zhao et al., 2021), it would be beneficial for future work to experiment with other instruction variants.

**API in Context**  Including the API in context improves performance in the Zero-shot case, but not in the Few-shot case. In the Max-shot (shown in Table 3) and Max-shot+Instruction experiments, including the API in context *harms* overall performance. This is likely because the API has a large number of tokens. Much less data can be included in the prompt with this method, and some in-context examples may need to be removed from the prompt entirely.

**Dynamic Prompting**  Dynamic prompting is incredibly effective, causing at least a 40% relative improvement in all 4 metrics. Despite this performance improvement, the model still often makes mistakes where API elements are hallucinated. Since prompting with the full API directly seems to harm performance, solving the hallucination of API elements under in-context learning is an important area for further study.

### 5.1  Manual Analysis

Our manual annotation indicates that although the finetuned T5 model outperforms GPT4+DP on the automated metrics, in-context learning with GPT-4 slightly outperforms the finetuned T5 model in

| Model | Standard Eval. | | | | Function-based Eval. | | | | Manual Eval. |
|---|---|---|---|---|---|---|---|---|---|
| | EM | ROUGE | F1 | BLEU | EM | ROUGE | F1 | BLEU | |
| GPT3.5 Turbo | 0.0 | 34.47 | 5.09 | 29.12 | 0.5 | 42.69 | 10.68 | 33.86 | – |
| GPT4 | 3.0 | 52.70 | 42.62 | 40.08 | 15.5 | 63.25 | 52.73 | 52.15 | – |
| GPT4+API | 1.50 | 43.18 | 40.96 | 31.24 | 12.00 | 54.33 | 48.09 | 41.95 | – |
| GPT4+DP | 15.00 | 69.59 | 61.83 | 62.60 | 41.50 | 80.80 | 73.56 | 74.91 | **56.91** |
| GPT4+API+DP | 15.00 | 63.99 | 57.83 | 57.47 | 36.00 | 74.78 | 67.94 | 68.71 | 50.79 |
| T5 (Finetuned) | **17.00** | **76.73** | **72.06** | **68.81** | **45.50** | **87.58** | **83.46** | **81.70** | 52.38 |

Table 3: Model performance in different evaluation settings. All models are evaluated under Max-shot in-context learning, except the finetuned T5 model. The full results on Zero-shot, Few-shot, and Max-shot+Instruction can be found in Appendix E. EM is normalized exact match. Settings with +API indicate where we append the description of the full OfficeScripts API to the prompts, and settings with +DP indicate where dynamic prompting is applied. In the standard evaluation setting we report metrics for the full prediction against the original gold standard output, while in the function-based evaluation setting we report metrics on normalized code predictions and outputs. Normalization replaces parameters, ranges, and numbers with placeholder values.

manual correctness judgments. Thus, although the automated metrics can help with rough comparisons, they do not give a complete picture of model performance.

We also perform a 2-sample, one-sided t-test to compare the value of each automated metric for manually-judged incorrect predictions vs. manually-judged correct predictions. This test has the alternative hypothesis that the metric has a lower average when the example is manually determined to be incorrect. We perform these tests for the 3 top-performing models and all metrics. We reject the null hypothesis with a p-value of 0.01 in all cases. These tests indicate a statistically significant difference between the values of automated metrics for correct vs. incorrect predictions, suggesting that automated metrics may serve as a convenient (though incomplete) replacement for metrics based on execution equivalence or manual judgment.

We also perform a qualitatitive exploration of remaining error types. About 20% of the annotated examples simply do not compile for the top three scoring models, often because they hallucinate API elements or formulas. Among the examples that compile correctly, we identify three major remaining error types for the finetuned T5 model: misunderstanding the user's intention, targeting the wrong object or incorrect cell range, and accomplishing the correct task in a different way than intended. Full examples of these failure modes are shown in Figure 5. We also identify major error types for the GPT4+DP model, finding that misunderstanding the users' intentions/solving the wrong task and targeting the wrong object or cell

range are also the most prevalent. Examples of errors made by GPT4+DP are included in Table 8 of Appendix F. In contrast to the T5 model, the GPT4+DP model has more errors where a completely different task is solved than what is queried. This is a potential artifact of in-context learning that should be addressed. Both models suffer from overwriting important data, as shown in the final example of Table 8.

## 6 Case Study: Formatting Rules in Excel

The INSTRUCTEXCEL benchmark contains a wide range of Excel tasks, including charting/plotting (Lee et al., 2021), formatting (Singh et al., 2022), and formulas (Chen et al., 2021b). We can derive specific benchmarks for these individual downstream tasks from INSTRUCTEXCEL. In this case study, we present an analysis of conditional formatting rules as a downstream task.

To focus on the task of learning conditional formatting rules, we extract all examples from the INSTRUCTEXCEL dataset that involve any formatting operator. These extracted samples are then manually validated to ensure their relevance to formatting tasks. We obtain a total of 660 valid CF tasks after filtering.

To facilitate benchmarking, we transform the extracted conditional formatting tasks into an Intermediate Language (IL) representation specifically designed for formatting operators (Singh et al., 2022). Figure 6 showcases an example of this IL.

To evaluate the performance of various approaches on this task, we employed the original natural language (NL) descriptions associated with

| Error | Ex. Input | Pred. Output | Gold Std. |
|---|---|---|---|
| User intention misunderstood | Change Chart title on sheet data | chart.getTitle().setVisible(true); | chart.getTitle().setText("Chart Title1"); |
| Incorrect target for data operation | Apply gridline on sheet1 | selectedSheet.getRange("A1:D1").getFormat().setGridlines(true); | selectedSheet.setShowGridlines(true); |
| Two ways of accomplishing same task | Merge and center first row on sheet1 | selectedSheet.getRange("1:1").merge(false); selectedSheet.getRange("1:1").getFormat().setHorizontalAlignment(ExcelScript.HorizontalAlignment.center); | selectedSheet.getRange("A1:J1").merge(false); selectedSheet.getRange("A1:J1").getFormat().setHorizontalAlignment(ExcelScript.HorizontalAlignment.center); |

Figure 5: Examples of errors made by the finetuned T5 model on test set examples. Each line shows an example of a common type of error.

**Query:**
"Highlight Negative Value in P/L Column. Apply conditional formatting on sheet 1"

**Excel Script:**

```
function main(workbook: ExcelScript.Workbook) {
    let conditionalFormatting: ExcelScript.ConditionalFormat;
    let selectedSheet = workbook.getActiveWorksheet();
    // Change cell value from range E2:E189 on selectedSheet
    conditionalFormatting = selectedSheet.getRange("E2:E189")
        .addConditionalFormat(ExcelScript.ConditionalFormatType.cellValue);
    conditionalFormatting.getCellValue().getFormat().getFont().setColor("#9C5700");
    conditionalFormatting.getCellValue().getFormat().getFill().setColor("#FFEB9C");
    conditionalFormatting.getCellValue().setRule({
        formula1:"=10",
        formula2:undefined,
        operator:ExcelScript.ConditionalCellValueOperator.greaterThan,});
}
```

**Formatting Rule:**
GreaterThan("E2:E189", "10")

Figure 6: Example of a CF task from INSTRUCTEXCEL and the associated Intermediate Language (IL) rule.

| Model | Setup | EM | SM | ExM |
|---|---|---|---|---|
| GPT 3.5 | Zero-shot | 2.3 | 5.6 | 12.1 |
| | Few-shot (3) | 42.3 | 42.2 | 64.3 |
| | Max-shot | 50.8 | 59.8 | 67.2 |
| GPT 4 | Zero-shot | 3.7 | 6.8 | 14.2 |
| | Few-shot (3) | 45.6 | 60.5 | 70.2 |
| | Max-shot | 53.5 | 63.4 | 73.3 |

Table 4: Baseline results on the CF task extracted from INSTRUCTEXCEL. EM, SM and ExM indicate Exact Match, Sketch Match and Execution Match respectively.

the tasks to generate simplified formatting rules. As these rules were already in a parsed format, we report the following evaluation metrics: (1) *Exact Match*, (2) *Sketch Match* over the AST of the parsed rules, and (3) *Execution Match* - Verifying the equivalence of the resulting formatting by executing the generated rules on data and comparing the outcomes (Singh et al., 2022, 2023).

Table 4 illustrates the performance of various baselines on the task of learning conditional formatting rules. GPT-4 with Max-shot in-context learning achieves the highest performance among the evaluated approaches. Our analysis showcases the feasibility of extracting relevant data, transforming it into an intermediate format, and benchmarking various approaches for the specific task of conditional formatting. These findings serve as a compelling motivation for the broader usage of the INSTRUCTEXCEL dataset in advancing research and development within the Excel domain.

## 7 Conclusion

We introduce INSTRUCTEXCEL, a benchmark to investigate the capability of models in generating Excel OfficeScript code from NL instructions. IN-STRUCTEXCEL consists of over 10,000 samples covering 170+ Excel operations across 2,000 pub-

licly available Excel spreadsheets. Our experimental results show that (1) GPT-4 (2) more in-context examples, and (3) dynamic prompting help improve performance on this benchmark, while more-detailed in-context instructions and including the API description do not help. Finetuning aids performance on automated metrics, though our manual annotation shows that the difference between finetuning and in-context learning is not conclusive. INSTRUCTEXCEL is still a challenging benchmark for state of the art models like GPT-4 and T5. We also demonstrate, through a case study on conditional formatting, that INSTRUC-TEXCEL can be used as source data for more specific study of important sub-tasks in Excel. We hope that improvements on INSTRUCTEXCEL will enable novice users to perform complicated actions in Excel quickly, boosting productivity and teaching important Excel functionality in the process.

## 8 Limitations

Our focus in INSTRUCTEXCEL is on instructions written in English. A typical non-expert user workflow often involves languages other than English. The limitation to not include local language in IN-STRUCTEXCEL will be addressed in future work. The programming language under study, Excel OfficeScripts, is specific to Microsoft Excel, so our

results may not generalize to other spreadsheet software. Our focus in this work is to release a targeted benchmark which, if performance is improved, will enable a core set of NL to code functionality in Microsoft Excel. The potential risks from this work are minimal. Excel OfficeScripts is a language designed to be secure, targeted only at manipulating data inside Excel spreadsheets. Therefore, there is little risk that a model will produce overtly harmful code in this domain.

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

## A Full Schema Example

Table 5 shows multiple input-output examples. The examples include before and after images of the relevant regions of the Excel spreadsheet.

## B HIT Screenshot

Figure 7 shows a screenshot of the interface shown to crowdworkers while inputting file descriptions, NL instructions, and copy-pasted code from the Automate feature.

## C Longer Instruction

The more detailed instruction used in the Max-shot+Instruction setting is: '*Generate a function with excel script to execute the action given below in NL. You also need to generate comment describing the operation you are performing. Make sure to generate a valid excel operation and pass appropriate parameters as provided in the action information. Simple solution is preferred over a complex one.*'

## D Finetuning Parameters

T5 training was done for $5,000$ steps, with learning rate increasing linearly from $0$ to $1 \cdot 10^{-4}$ for the first $1,000$ steps, and then linearly down to $0$ for the remaining steps. We restrict both the input and output size to $1,000$ tokens. If inputs and outputs are denoted as $I_i$ and $O_i = \{t_k^{(i)}\}_{k=1}^{|O_i|}$ respectively, then the finetuning objective is to maximize the probability of the true output given the input,

$$p(O_i|I_i) = \prod_{k=1}^{|O_i|} p(t_k^{(i)}|t_1^{(i)}, \ldots t_{k-1}^{(i)}, I_i) \ .$$

At inference time, we use beam search with beam size $5$, and select the top $1$ completion for final evaluation.

## E Full Experimental Results

The full results for the Zero-shot, Few-shot, Max-shot, and Max-shot+Instruction settings on all models are reported in Table 6. In addition to the evaluation metrics reported in Table 6, we also report the same metrics after replacing all parameters, ranges, and numbers in both the gold standard and predicted outputs with placeholder values (function-based evaluation). These modified metrics are reported in Table 7. Although the metrics are much higher in this setting, the same overall trends hold.

## F Failure modes of GPT4+DP

Table 8 shows major error modes for GPT4+DP. These error modes are similar to those seen for T5 in Figure 5.

| Query | Gold Standard Output |
|---|---|
| Find unique from column Method range C11:C42 on G17 on sheet Au | ```function main(workbook: ExcelScript.Workbook) {``` 
 ```let selectedSheet = workbook.getActiveWorksheet();``` 
 ```// Set range G17 on selectedSheet``` 
 ```selectedSheet.getRange("E17").setFormulaLocal(``` 
 ```    "=unique(C11:C42)");}``` |

| FA | MS |
|---|---|
| NAA | |
| FA | AAS |
| FA | AAS |
| FA | AAS |
| PR,AR | MS |
| FA | ES |
| FA | ICP |
| FA | ES |

| FA | MS | |
|---|---|---|
| NAA | | |
| FA | AAS | |
| FA | AAS | FA |
| FA | AAS | NAA |
| PR,AR | MS | PR,AR |
| FA | ES | AR |
| FA | ICP | BLEG |
| FA | ES | |

| Query | Gold Standard Output |
|---|---|
| Find top 2 value from range B6:P22 on sheet CVs | ```function main(workbook: ExcelScript.Workbook) {``` 
 ```let conditionalFormatting: ExcelScript.ConditionalFormat;``` 
 ```let selectedSheet = workbook.getActiveWorksheet();``` 
 ```// Change top bottom from range B6:P22 on selectedSheet``` 
 ```conditionalFormatting = selectedSheet.getRange("B6:P22")``` 
 ```    .addConditionalFormat(``` 
 ```      ExcelScript.ConditionalFormatType.topBottom);``` 
 ```conditionalFormatting.getTopBottom().getFormat()``` 
 ```    .getFont().setColor("#9C0006");``` 
 ```conditionalFormatting.getTopBottom().getFormat()``` 
 ```    .getFill().setColor("#FFC7CE");``` 
 ```conditionalFormatting.getTopBottom().setRule(``` 
 ```    {rank:2,``` 
 ```    type:ExcelScript.ConditionalTopBottomCriterionType.topItems,});}``` |

| ** 0.00 | ** 0.00 | 98.65 |
|---|---|---|
| ** 98.84 | ** 96.18 | ** 0.00 |
| 49.24 | 66.58 | ** 0.00 |

| ** 0.00 | ** 0.00 | 98.65 |
|---|---|---|
| ** 98.84 | ** 96.18 | ** 0.00 |
| 49.24 | 66.58 | ** 0.00 |

| Query | Gold Standard Output |
|---|---|
| replace all N/A with 0 | ```function main(workbook: ExcelScript.Workbook) {``` 
 ```let selectedSheet = workbook.getActiveWorksheet();``` 
 ```selectedSheet.replaceAll("N/A", "0",``` 
 ```    {completeMatch: false, matchCase: false});}``` |

| Query | Gold Standard Output |
|---|---|
| replace all N/A with 0 | ```function main(workbook: ExcelScript.Workbook) {``` 
 ```let selectedSheet = workbook.getActiveWorksheet();``` 
 ```selectedSheet.getRange("B4:B10").replaceAll("N/A", "0",``` 
 ```    {completeMatch: false, matchCase: false});}``` |

Table 5: Example natural language queries and gold standard code outputs. Examples $1-3$ include images of the Excel spreadsheet before and after executing the gold standard code. Example 4 shows the same query on the same data as example 3, solving using slightly different code.

Figure 7: Screenshot of the HIT given to crowdworkers who created the INSTRUCTEXCEL dataset.

| Model | Setup | EM | ROUGE | F1 | SacreBLEU |
|---|---|---|---|---|---|
| GPT 3.5 Turbo | Zero-shot | 0.0 | 29.46 | 5.44 | 21.75 |
| | Few-shot (3) | 1.0 | 34.94 | 11.10 | 28.31 |
| | Max-shot | 0.0 | 34.47 | 5.09 | 29.12 |
| | Max-shot+Instruction | 0.0 | 35.67 | 2.67 | 29.53 |
| GPT 4 | Zero-shot | 0.0 | 21.59 | 6.75 | 16.90 |
| | Few-shot (3) | 1.5 | 42.57 | 31.73 | 31.09 |
| | Max-shot | 3.0 | 52.70 | 42.62 | 40.08 |
| | Max-shot+Instruction | 3.0 | 41.86 | 38.34 | 31.71 |
| GPT 4 + API | Zero-shot | 0.00 | 30.57 | 10.66 | 20.32 |
| | Few-shot | 3.50 | 41.79 | 31.63 | 30.56 |
| | Max-shot | 1.50 | 43.18 | 40.96 | 31.24 |
| | Max-shot+Instruction | 1.50 | 33.15 | 30.95 | 23.80 |
| GPT 4 + DP | Zero-shot | 0.00 | 22.84 | 7.00 | 18.16 |
| | Few-shot | 11.00 | 67.28 | 58.23 | 59.23 |
| | Max-shot | 15.00 | 69.59 | 61.83 | 62.60 |
| | Max-shot+Instruction | 15.00 | 69.22 | 61.95 | 62.29 |
| GPT 4 + API + DP | Zero-shot | 0.00 | 30.37 | 10.28 | 20.15 |
| | Few-shot | 11.00 | 62.88 | 55.05 | 55.71 |
| | Max-shot | 15.00 | 63.99 | 57.83 | 57.47 |
| | Max-shot+Instruction | 13.50 | 64.61 | 60.15 | 57.94 |
| T5 | Finetuning | 17.00 | 76.73 | 72.06 | 68.81 |

Table 6: Model performance in different evaluation settings. EM is normalized exact match.

| Model | Setup | EM | ROUGE | F1 | SacreBLEU |
|---|---|---|---|---|---|
| GPT 3.5 Turbo | Zero-shot | 0.0 | 36.67 | 4.13 | 25.65 |
| | Few-shot (3) | 1.0 | 42.02 | 11.23 | 34.55 |
| | Max-shot | 0.5 | 42.69 | 10.68 | 33.86 |
| | Max-shot+Instruction | 0.5 | 44.04 | 6.64 | 35.59 |
| GPT 4 | Zero-shot | 0.0 | 26.81 | 5.77 | 22.56 |
| | Few-shot (3) | 1.5 | 50.05 | 35.35 | 39.25 |
| | Max-shot | 15.5 | 63.25 | 52.73 | 52.15 |
| | Max-shot+Instruction | 7.0 | 52.10 | 45.23 | 40.57 |
| GPT 4 + API | Zero-shot | 0.00 | 35.06 | 9.90 | 23.49 |
| | Few-shot | 3.50 | 49.79 | 35.75 | 38.31 |
| | Max-shot | 12.00 | 54.33 | 48.09 | 41.95 |
| | Max-shot+Instruction | 1.50 | 43.86 | 36.08 | 32.28 |
| GPT 4 + Dynamic Prompt | Zero-shot | 0.00 | 28.26 | 6.88 | 22.97 |
| | Few-shot | 33.50 | 77.91 | 68.91 | 71.19 |
| | Max-shot | 41.50 | 80.80 | 73.56 | 74.91 |
| | Max-shot+Instruction | 41.00 | 80.12 | 73.30 | 74.34 |
| GPT 4 + API + Dynamic Prompting | Zero-shot | 0.00 | 34.91 | 9.96 | 23.22 |
| | Few-shot | 32.00 | 73.47 | 64.68 | 66.86 |
| | Max-shot | 36.00 | 74.78 | 67.94 | 68.71 |
| | Max-shot+Instruction | 38.00 | 76.09 | 70.81 | 70.03 |
| T5 | Finetuning | 45.50 | 87.58 | 83.46 | 81.70 |

Table 7: Results ignoring parameters, ranges, and numbers (function-based evaluation).

| Error | Ex. Input | Pred. Output | Gold Std. |
|---|---|---|---|
| User intention misunder-stood/incorrect task | Change the legends position on sheet data in chart | ```
selectedSheet.addChart(
ExcelScript.ChartType.
columnClustered,
selectedSheet.getRange(
"A28:B30"));
``` | ```
chart_3.getLegend()
.setPosition(
ExcelScript.
ChartLegendPosition.top);
``` |
| Incorrect target for data opera-tion | Data Validation for drop down for column Lab from range A11:A101 on sheet Au | ```
selectedSheet
.getRange("B13:B16")
.getDataValidation()
.setRule(
{list:{inCellDropDown:true,
source:"=$A$11:$A$101"}});}
``` | ```
selectedSheet
.getRange("A11")
.getExtendedRange(
ExcelScript.
KeyboardDirection.down)
.getDataValidation()
.setRule({
list:{inCellDropDown:true,
source:"=$A$11:$A$101"}});}
``` |
| Incorrect for-mula applied | Apply proper function to cell C4 in sheet Supplementary Data 2 | ```
selectedSheet
.getRange("C4")
setFormula(
"=AVERAGE(C5:C281)");
``` | ```
selectedSheet
.getRange("I10")
.setFormulaLocal(
"=proper(C4)");
``` |
| Overwriting im-portant data | Apply isnumber function to cell B7 of column price of sheet 8.S1a Airline Demand | ```
selectedSheet
.getRange("B7")
.setFormulaLocal(
"=ISNUMBER(B7)");
``` | ```
selectedSheet
.getRange("H9")
.setFormulaLocal(
"=ISNUMBER(B7)");
``` |

Table 8: Types of errors made by the GPT4+DP model. The error type is listed, along with an NL query, predicted (incorrect) output by GPT4+DP, and the gold standard output. Code is shortened to only include the relevant, erroneous function calls.