# OpenReview forum: "InstructExcel: A Benchmark for Natural Language Instruction in Excel"
_EMNLP/2023/Conference — EMNLP 2023 Findings_

### Official Review · Reviewer_em7S · 2023-07-24

**Soundness:** 4

**Excitement:**

3: Ambivalent: It has merits (e.g., it reports state-of-the-art results, the idea is nice), but there are key weaknesses (e.g., it describes incremental work), and it can significantly benefit from another round of revision. However, I won't object to accepting it if my co-reviewers champion it.

**Paper Topic And Main Contributions:**

The paper contributes a code generation dataset for Excel: the input is a natural language instruction and the output is a typescript representing a series of Excel operations. The dataset contains 10k samples over 2,000 publicly available spreadsheets. Preliminary experiments show that GPT-4 performs poorly, indicating plenty of room for improvement. The authors also ran a series of baseline approaches to improve the performance, such as increasing in-context examples, dynamic prompting etc.

**Questions For The Authors:**

A. What is the "inter-annotator agreement rate"? Give a natural language description to another worker, how often can the other worker reproduce the same action? I believe such a measurement is important, since it is easy for an annotator to think that the description they came up with is unambiguous, while another annotator interpreted it differently.

B. How did you make sure that the input descriptions are diverse and of high quality? Asking annotators to come up with their own description is different from the distribution of descriptions users would naturally generate. While I understand that it'd be difficult to collect data from the real users, providing more details in this respect in the main paper would be helpful.

**Reasons To Accept:**

- The paper provides a useful dataset for the community (instruction -> code);
- The idea of "recording annotator's action' to form the annotated output is interesting and was not widely used in the semantic parsing literature before.
- The empirical results (e.g. detailed instructions do not help much) are interesting to the community.

**Reasons To Reject:**

It'd be useful for the authors to provide further details on how they controlled the quality of the dataset. On the other hand I would not set a high bar for the inter-annotator agreement rate and the justification for diversity as long as they are reasonable and reported in the paper. I'm ready to raise my score on soundness.

P.S. Sorry that I don't have time to go through the details in the appendix if you provide those details there (and I believe that the main paper should be self-contained). Feel free to address them in your updated version when an additional page is available for camera ready.

**Reproducibility:**

4: Could mostly reproduce the results, but there may be some variation because of sample variance or minor variations in their interpretation of the protocol or method.

**Reviewer Confidence:**

4: Quite sure. I tried to check the important points carefully. It's unlikely, though conceivable, that I missed something that should affect my ratings.

**Typos Grammar Style And Presentation Improvements:**

L475: while I believe that GPT-4 is larger than GPT-3.5 and larger models are probably indeed better, the fact that GPT-4 performance is better than GPT-3.5 does not imply that larger models are better because they are blackbox APIs and can differ on many different axes (e.g., training data). You could draw the conclusion if you compared fine-tuned T5-large vs T5-medium (and many other T5-series) though.

---

> ### Author Rebuttal · Authors · 2023-08-29
>
> We are thankful to the reviewer for their encouraging feedback. We appreciate the reviewer's judgment that we provide a “useful dataset” with the “interesting” “idea of ‘recording annotator's action' to form the annotated output”. We address the reviewer’s concerns below.
>
> R3.1 “inter-annotator agreement rate”: This is an interesting question, thank you.
>
> We ran a qualitative analysis on inputs which appear at least twice in the benchmark. We find that most of these inputs have very high “inter-annotator agreement rate”; for example, there is only one way to satisfy “freeze row 1”: “selectedSheet.getFreezePanes().freezeAt(selectedSheet.getRange("1:1"));”. However, some of these inputs can be satisfied multiple ways, for example, we see “replace all N/A with 0” satisfied by both “selectedSheet.replaceAll("N/A", "0", {completeMatch: false, matchCase: false});” and “selectedSheet.getRange("B4:B10").replaceAll("N/A", "0", {completeMatch: false, matchCase: false});”. We can include this qualitative analysis in the final version of the paper, as well as a report on the average value of the automated metrics computed pairwise within each query with at least 2 completions.
>
> R3.2: “details on how they controlled the quality of the dataset.”, “input descriptions are diverse and of high quality”: Thanks for the great question. Here are more details on our process to control data quality.
>
> 1. Multiple rounds of iterations over the instructions with pilot studies
> 2. Various examples provided to annotators to demonstrate a variety of actions (these could be toggled on and off in the annotation interface)
> 3. Instructed annotators to not overuse or underuse certain sets of actions after observing the distribution of actions in pilot rounds
> 4. Used annotators that are paid hourly ($12 USD per hour) for delivering high quality work and not annotators that are motivated to click on as many hits as possible in a short span of time
> 5. Employed an annotation project manager to facilitate training of annotation team, and to facilitate later communication with team
> 6. Inspected sample annotations
> 7. Selection of a certain set of files that were most suitable for the task: large enough to have nontrivial contents, but not so large that it would be impossible for annotators to understand the content (files between 60KB and 100KB)
> 8. Selected only English-language files
> 9. Filtered for password and corrupted files
>
> We emphasized getting diverse and high quality descriptions during our selection of annotation vendors and in our pilot studies. Our initial pilot study indeed had repetitive Excel operations which we had to reject. We requested our vendor to arrange a pool of annotators who have basic familiarity with Excel applications. We also provided the vendor a list of top popular Excel operations which they must use. Additionally, we introduced an input field asking users to mention the intent behind the operation they want to perform (Figure 8). Inspired by other benchmark papers, we periodically reviewed the data and plotted the histogram of Excel operations (Figure 3), and accordingly informed the vendor of (1) the Excel operations they should not use anymore since they are already overpopulated, and (2) the operations they are encouraged to use since they are missing/underpopulated. The 170+ operations our benchmark contains cover diverse use cases of the Excel software, with the popular use cases dominant in the distribution (Figure 3).
>
> Based on the reviewer's suggestion, we will use our extra page to bring content from the appendix to the main paper in the camera ready version.
>
> R3.3.: Typos Grammar Style And Presentation Improvements: Thanks for pointing this out, we will update this in the camera ready version.

---

### Official Review · Reviewer_6cHD · 2023-08-04

**Soundness:** 4

**Excitement:**

3: Ambivalent: It has merits (e.g., it reports state-of-the-art results, the idea is nice), but there are key weaknesses (e.g., it describes incremental work), and it can significantly benefit from another round of revision. However, I won't object to accepting it if my co-reviewers champion it.

**Missing References:**

*I'm not familiar with this line of work.*

**Paper Topic And Main Contributions:**

The authors share a benchmark dataset that contains (1) links to ~2000 public spreadsheets, (2) instructions for ~10K sample actions that may be performed in those spreadsheets, as queries, and (3) OfficeScript code that executes those actions, as target labels, (obtained via the Automate feature on Excel which records annotator actions into code).

The authors perform simple baseline experiments to identify the efficacy of common approaches: (1) zero-shot, few-shot, and *dynamic* [1] prompting on GPT 3.5 & 4, and (2) fine-tuning T5 large models.

The authors also highlight the potential to derive fine-grained benchmarks using subsets of the proposed dataset, providing a case study for the popular task of conditional formatting rules.

[1] Liu 2021, What Makes Good In-Context Examples for GPT-3?

**Reasons To Accept:**

- The authors provide a challenging benchmark for the important task of (generating code for) automatically solving spreadsheet tasks via natural language instructions.
- The authors perform experiments on common methods which shed light on the capabilities of current popular models/approaches and can serve as baselines for future research.
- Writing is overall good.

**Reasons To Reject:**

- The use of traditional metrics based on distance to golden labels may not be a suitable evaluation metric for this task. The justification proposed by the authors in Appendix D does not convince me otherwise, as the hypothesis for the t-test is very weak (i.e., that the average scores of the automated metrics are lower for incorrect samples from a single experiment, i.e., GPT-4 with max-shot prompting). This does not justify the use of the proposed metrics *for comparing the performance between different models*. A better alternative may be to show the *true* accuracy determined by human annotators based on random-selected samples for *each* main baseline method, alongside the scores from automated metrics. In fact, there is no mention of true model accuracy, i.e., the portion of *acceptable* answers, determined by annotators.
  - For this, my soundness score for the baseline experiment part would be 2/5. If the authors could show an automated metric that can faithfully represent the ability of models to solve the task, that would be a highly relevant contribution alongside the proposed benchmark. However, the paper seems to fall short of this in its current state. Other than the experiments, the benchmark itself looks sound.
- It may be informative to show the statistics for the average number of examples used in 3-shot and max-shot experiments, accounting for truncation. I'm also concerned about the truncation of *table data* due to limited context sizes discussed in Line ~382, as this would be very detrimental to task performance. While I understand that this is not a trivial issue to solve, some discussion of the severity of data truncation or statistics may help readers consider this when reading the experimental results. While the exact truncation protocol is explained in a dedicated subsection (4.3), It seems that the consequences of truncation have not been thoroughly considered by the authors. *If data truncation does not affect a significant number of samples, it would be nice if this is mentioned.

**Reproducibility:**

4: Could mostly reproduce the results, but there may be some variation because of sample variance or minor variations in their interpretation of the protocol or method.

**Reviewer Confidence:**

3: Pretty sure, but there's a chance I missed something. Although I have a good feel for this area in general, I did not carefully check the paper's details, e.g., the math, experimental design, or novelty.

**Typos Grammar Style And Presentation Improvements:**

- The results section could be expanded to more effectively deliver key insights on baselines. E.g., the expanded performance table (Table 5) could be moved to the main text, with a more thorough summary of findings in the main text of Section 5. Some content such as experimental details and Figure 3 could be shortened.
- Comprehensive examples figures for selected samples from the dataset may help the reader intuitively understand the content of the benchmark. I.e., it may be better to organize the input, output (from Figure 6), and data contents (Figure 7) of each sample into a single, well-organized, readable figure. It would also be nice to show *multiple*, say 3, of such figures, to show a wide range of samples.

---

> ### Author Rebuttal · Authors · 2023-08-29
>
> We are grateful to the reviewer for acknowledging our work to contain a “challenging benchmark” and “overall good writing”. We address the two major concerns below:
>
> R2.1: “ traditional metrics based on distance to golden labels may not be a suitable evaluation… A better alternative may be to show the true accuracy….”: This was a very fruitful question, thank you. We manually annotated 40 examples for the bottom 3 experimental settings in Table 2: GPT4+DP, GPT4+API+DP, and Finetuned T5. For this manual annotation, we download the associated spreadsheets and compare the results of executing the predicted code against the natural language queries. We find that among these 40 annotated examples, GPT4+DP shows 65% accuracy, GPT4+API+DP shows 50% accuracy, and the Finetuned T5 model shows 57.5% accuracy. This small-scale analysis suggests that you are absolutely correct; although the automated metrics are useful coarse-grained measures of model accuracy, they cannot replace manual annotation for detailed model comparison. We intend to extend the manual annotation to all 200 test examples and report these results in the final version of the paper. This nuance of evaluation will be very important information for future users of our benchmark.
>
> Automation of this process remains out of scope for the current work, as it is a non-trivial problem. It may be possible to use PowerAutomate to execute the predicted and gold standard code completions, and then load them into Python using Pandas to compare the results. However, it is unclear how to execute this workflow programmatically (search for “run PowerAutomate from command line” to see others struggling with the same problem). We will include more discussion of the difficulties of automated evaluation in the final version.
>
> R2:2: “average number of examples used in 3-shot and max-shot experiments, accounting for truncation. ”:  Given that our main contribution is the benchmark itself, our data truncation scheme was intended to be a baseline for future work to build upon. We analyzed the two most space-restricted settings in our experiments, the maxshot+instruction settings with both the GPT3.5 model with 16k context size, and the GPT4 model with 32k context size + API in the prompt. Both settings always include the intended 10 in-context examples in the prompt, and the data included in the examples retain an average of 19.36 lines in the first case and 15.36 lines in the second case. This typically guarantees that the headers and some data are included from the first spreadsheet in the Excel file, though occasionally later spreadsheets have important data as well. We hope that follow-up work will address this non-trivial issue. There has been a recent trend towards increasing input context length, so perhaps increased context size will make this problem irrelevant in the future.
>
> R2.3: Typos Grammar Style And Presentation Improvements: Thank you for your comments on improving the presentation of the experiments and examples. We will certainly implement these suggestions in the final version.

---

### Official Review · Reviewer_DQNq · 2023-08-12

**Paper Topic And Main Contributions:** 1. **Performance Metrics**
**Soundness:** 4

**Excitement:**

3: Ambivalent: It has merits (e.g., it reports state-of-the-art results, the idea is nice), but there are key weaknesses (e.g., it describes incremental work), and it can significantly benefit from another round of revision. However, I won't object to accepting it if my co-reviewers champion it.

**Questions For The Authors:**

**A.** The manual annotation process revealed some disagreements among experts on the correctness of model outputs. How do you plan to refine the evaluation criteria to ensure more clarity and consistency in future iterations of the benchmark?

**B.** While the benchmark focuses on Excel, many other software applications could benefit from NLP integration. Do you have plans to expand or adapt the benchmark to other software applications or platforms?

**C.** How does the INSTRUCT EXCEL benchmark compare to other Code based instruction benchmarks in terms of complexity and diversity of tasks? Can you provide insights into its uniqueness or advantages over other benchmarks?

**D.** Can you elaborate on the real-world applicability of the benchmark? How does performance on this benchmark translate to effectiveness in real-world Excel tasks?

**Reasons To Accept:**

1. The paper introduces "INSTRUCT EXCEL", a systematic benchmark for evaluating language models' interaction with Microsoft Excel.
2. It provides a broad evaluation spectrum, testing various models like GPT and T5 across setups such as zero-shot, few-shot, and max-shot.
3. A novel function-based evaluation approach is introduced, emphasizing models' functional understanding over exact output matching.
4. The research bridges language models and spreadsheet software, heralding a new research domain with potential real-world applications in industries relying on tools like Excel.
5. The study offers insights into the intricacies of manual annotation, underscoring the importance of collaborative efforts in clarifying output ambiguities and enhancing model evaluations.

**Reasons To Reject:**

1. **Ambiguity in Evaluation**: The manual annotation process, as discussed in the paper, indicates that even experts found it challenging to agree on the correctness of model outputs. This could raise questions about the consistency and clarity of evaluation criteria.

2. **Limited Scope**: While the focus on Excel is pertinent given its widespread use, the benchmark might be considered too niche. Some might argue for a more generalized benchmark that encompasses a broader range of software applications beyond Excel.

3. **Lack of Comparative Analysis**: While the paper introduces a new benchmark and tests several models on it, there might be limited discussion on how these results compare to other existing benchmarks or real-world tasks, potentially reducing its impact.

4. **Potential Overemphasis on Metrics**: While metrics like EM, ROUGE, and SacreBLEU are valuable, the paper might be critiqued for not diving deeper into qualitative insights or providing more detailed error analyses.

**Reproducibility:**

5: Could easily reproduce the results.

**Reviewer Confidence:**

4: Quite sure. I tried to check the important points carefully. It's unlikely, though conceivable, that I missed something that should affect my ratings.

---

> ### Author Rebuttal · Authors · 2023-08-29
>
> We appreciate the reviewer’s detailed and insightful feedback. We are encouraged by the reviewer’s acknowledgment of our contributions: “Performance Metrics”, “Model Setups”, “Function-Based Evaluation”, “Manual Annotation” and “Visual Aids”. We are motivated that the reviewer finds InstructExcel to be a “systematic benchmark”, appreciates the “broad evaluation spectrum” and “novel function-based evaluation approach”, and suggests our “research bridges language models and spreadsheet software, heralding a new research domain with potential real-world applications in industries” and “offers insights into the intricacies of manual annotation”.
>
> We respond to major comments below:
>
> R1.1: “Ambiguity in Evaluation”: Excel is used by billions of users with diverse backgrounds consisting of experts and non-expert users. We selected annotators with diverse Excel skill levels to match the real world user distribution. Some of the paper's authors had extensive experience with OfficeScripts, while others had less experience with OfficeScripts (but more experience in training and evaluating LLMs). The less-experienced annotators looked for similarity of the model output with the gold answer, whereas the expert annotators allowed differences with the gold output as long as the API generated by the model compiles and does the required task. We can clarify this in the camera ready version. We will also describe two different guidelines for expert and less-experienced annotators.
>
> R1.2: “Limited Scope”: Our decision to focus on Excel is driven by the fact that Excel has over a billion users, and is still a hard task for state of the art models like GPT4. Excel workflow is part of daily life for many users and we would like to apply advancements in large language models to significantly improve their workflow. InstructExcel is a stepping stone in that direction. We hope the community will expand our benchmark to other tools in future.
>
> R1.3: “Lack of Comparative Analysis”, “real world excel tasks”: Our work is at the intersection of instruction learning and program synthesis. In the instruction learning space, the input is a natural language instruction and the output is an answer, story, poem, email etc. Natural Instructions [1], Supernatural Instructions [2], and Promptsource [3] are some popular instruction learning benchmarks, however they are focused on general NLP. Program synthesis benchmarks such as MBPP [4], APPS [5], CodeContests [6], CoNaLa [7], DJANGO [8], and HumanEval [9] on the other hand involve conversion of natural language to code, but this line of work has been predominantly focused on general purpose programming languages such as Python, and C++. Our benchmark InstructExcel instead focuses on Excel OfficeScripts, a TypeScript API which has not been explored in existing benchmarks.  We will ensure all relevant instruction learning and program synthesis benchmarks are included in the related work section.
>
> Our work can be immediately applied to a product used by billions of users, and our benchmark is based on real-world Excel files which provides unique practical advantages over other academic benchmarks on instruction following. Additionally, our task is more complex for average humans unlike conventional instruction following tasks such as writing a story, email, or joke. In addition to the main NL-to-code task, InstructExcel provides a reservoir for studying several other tasks e.g. the conditional formatting task, sub-task extraction (L083-092). During the creation of our benchmark, we emphasized usage of popular Excel operations. We worked directly with the vendor (Universal Human Relevance System) to provide diverse examples to the annotators, and train the annotators to create a wide distribution of realistic NL queries. We also introduced an additional field to further explain the users’ intentions (Figure 8). The 170+ operations used in our benchmark cover diverse use cases of the Excel software, with popular use cases dominant in the distribution (Figure 3).
>
> R1.4: “Potential Overemphasis on Metrics”, “detailed error analyses”: We conducted error analysis and qualitative evaluation. Due to space constraints, we moved this content to Appendices D and E. We will bring some of these to the main paper in the camera ready version upon getting additional space. If you would like to see additional analysis, please let us know what you suggest and we will incorporate it.
>
> References:
>
> [1] Mishra et al. 2022, Cross-task generalization via natural language crowdsourcing instructions.
>
> [2] Mishra et al. 2022, Super-naturalInstructions: Generalization via declarative instructions on 1600+ NLP tasks.
>
> [3] Bach et al. 2023, PromptSource: An integrated development environment and repository for natural language prompts.
>
> [4] Austin et al. 2021, Program synthesis with large language models.
>
> [5] Hendrycks et al. 2021, Measuring coding challenge competence with APPS.
>
> [6] Li et al. 2022, Competition-level code generation with AlphaCode.
>
> [7] Yin et al. 2018, Learning to mine aligned code and natural language pairs from Stack Overflow.
>
> [8] Oda et al. 2015, Learning to generate pseudo-code from source code using statistical machine translation.
>
> [9] Chen et al. 2021, Evaluating large language models trained on code.

---

### Meta-Review · Senior_Area_Chairs · 2023-10-04

**Recommendation:** 3

**Metareview:**

The paper "INSTRUCT EXCEL: A Benchmark for Evaluating Language Models in Spreadsheet Software" has received positive feedback for its innovative benchmark dataset and the insights it provides into language model performance. However, concerns about evaluation criteria, limited scope, and the need for further details on dataset quality control and inter-annotator agreement need to be addressed. The average soundness score is strong (4), indicating a solid foundation, while the average excitement score is ambivalent (3), suggesting room for improvement.

---

### Decision · Program_Chairs · 2023-10-07

**Decision:**

Accept-Findings

**Comment:**

The paper "INSTRUCT EXCEL: A Benchmark for Evaluating Language Models in Spreadsheet Software" has received positive feedback for its innovative benchmark dataset and the insights it provides into language model performance. However, concerns about evaluation criteria, limited scope, and the need for further details on dataset quality control and inter-annotator agreement need to be addressed. The average soundness score is strong (4), indicating a solid foundation, while the average excitement score is ambivalent (3), suggesting room for improvement.